# Evaluation of Prestress Loss Distribution during Pre-Tensioning and Post-Tensioning Using Long-Gauge Fiber Bragg Grating Sensors

**DOI:** 10.3390/s18124106

**Published:** 2018-11-23

**Authors:** Sheng Shen, Yao Wang, Sheng-Lan Ma, Di Huang, Zhi-Hong Wu, Xiao Guo

**Affiliations:** 1Department of Civil Engineering, Fuzhou University, Fuzhou 350108, China; N160527015@fzu.edu.cn; 2Hebei Province Key Laboratory of Evolution and Control of Mechanical Behavior in Traffic Engineering Structure, Shijiazhuang Tiedao University, Shijiazhuang 050043, China; 3CSCEC Strait Construction and Development Co., LTD, Fuzhou 350015, China; yao.wang@tom.com (Y.W.); francois@seu.edu.cn (Z.-H.W.); Xiao_Guo@tom.com (X.G.); 4Fujian Provincial Key Laboratory of Advanced Technology and Informatization in Civil Engineering, Fujian University of Technology, Fuzhou 350118, China; mashenglan@fjut.edu.cn

**Keywords:** prestress monitoring, prestress loss, pre-tensioning, post-tensioning, long-gauge fiber Bragg grating, strain distribution

## Abstract

Prestress loss evaluation in prestressed strands is essential for prestressed structures. However, the sensors installed outside the duct can only measure the total prestress loss. The sensors attached on strands inside the duct also have several problems, such as inadequate durability in an aggressive environment and vulnerability to damage during tensioning. This paper proposes a new installation method for long-gauge fiber Bragg grating (LFBG) sensors to prevent accidental damage. Then the itemized prestress losses were determined in each stage of the pre-tensioning and post-tensioning according to the LFBG measurements. We verified the applicability of the LFBG sensors for prestress monitoring and the accuracy of the proposed prestress loss calculation method during pre-tensioning and post-tensioning. In the pre-tensioning case, the calculated prestress losses had less deviation from the true losses than those obtained from foil-strain gauges, and the durability of the LFBG sensors was better than foil-strain gauges, whereas in post-tensioning case, the calculated prestress losses were close to those derived from theoretical predictions. Finally, we monitored prestress variation in the strand for 90 days. The itemized prestress losses at each stages of post-tensioning were obtained by the proposed calculation method to show the prospect of the LFBG sensors in practical evaluation.

## 1. Introduction

Prestressing of steel strands provides reversal stress to counteract in-service stress partially or entirely, improve the crack resistance, and reduce the deflection of prestressed structures. Thus, the tensile stress in steel strands can be maintained over time. However, the applied prestress may decrease gradually due to various reasons, such as the stress relaxation in the steel strands, concrete creep and shrinkage, friction between the strand and duct, and deformation of anchoring devices [1]. Moreover, long-term factors including aggressive environment, pitting, stress corrosion, and hydrogen embrittlement can decrease prestressing further and trigger a fracture of the strand that gives rise to accidents, causing the degradation of the nuclear containment vessel [2], the decrease in seismic performance of the concrete frame [3], and the collapse of a bridge [4]. Therefore, measuring and evaluating prestress loss of steel strand is imperative for maintenance and accurate assessment of prestressed structures.

Current sensing techniques for prestress loss measurement can be grouped into two categories: global measurements and local measurements. The global measurements are performed via elastomagnetic sensors [5,6,7], stress waves [8,9], Anchorage-Measurement-Access system [10,11], frequency [12,13], and modal parameters [14]. A common characteristic of the global measurement techniques is that the sensors are usually placed outside anchoring devices to obtain a “global” prestress of the strand. However, this approach has critical drawbacks. First, the “global” value fails to reflect the itemized prestress losses distributing along the strand. Moreover, most global measurements are indirect, complicated in data processing, and disturbed easily by electromagnetic interference. Last, the global measurement does not provide enough sensitivity to detect stress variations that may be quite small compared to the total stress of a prestressed strand due to micro cracks.

To overcome these disadvantages, researchers have focused on the local measurement that place the sensor on the surface of the strand to directly monitor prestresses at some pre-designated points. The local measurement is applied based on some electric sensors, such as strain gauges [15,16], piezoelectric transducer actuator [17] and so on. However, local measurement is more challenging to perform than global measurement due to the following three reasons. First, if the duct has a curved profile between the ends, the sensor or connecting line may be dislocated and damaged by the friction between the strand and duct during tensioning. Second, the interspace is small between the strand and duct. Some sensors, such as the elastomagnetic sensors, cannot be installed into the duct due to their size. The third reason is that grout used to fill the duct after tensioning may trigger a short circuit to the electric sensor without resin isolation.

The above challenges could be overcome using a fiber Bragg grating (FBG) sensor that is more suitable for long-term prestress monitoring than an electric sensor due to its small size, light weight, high stability, and durability. In recent years, FBG sensors have been widely used for dynamic strain-stress and vibration measurements in bridges [18,19,20], scour monitoring [21,22], reinforcement corrosion [23,24], and leakages in concrete structures and pipelines [25,26,27,28]. Two kinds of FBG sensors have been proposed for prestress monitoring. The first type is named as “smart strand” consisting of six helical wires and a core wire embedded an FBG sensor [29,30,31,32,33]. Although the FBG sensor can accurately measure prestress in the core wire, the “smart strand” also has two drawbacks in practice. First, for cost reduction, the position of each FBG sensor in a “smart strand” is predetermined in the production stage, and the distance between adjacent sensors is usually identical. However, these predetermined positions and distances may not match the required positions and distances in practical construction. Second, because the core wire is surrounded closely by six helical wires, it is difficult to connect the embedded FBG sensor in the core wire to the optical cable used to transmit the optical signal.

An improved “smart strand” type sensor has been proposed for monitoring prestress distribution by combining the Brillouin optical time domain analysis/refectory (BOTDA/R) sensor and the FBG sensor along a single optical fiber to solve this problem [34,35]. However, the measurement error of the BOTDA/R sensor was at least dozens of micro-strain, reducing the accuracy of the monitoring data. The results of Zhang’s experiments showed that the strain measurement error of AQ8603 (produced by Ando Electric Co. Ltd., Tokyo, Japan) based on the BOTDR technique was ±130 με (1.96 σ,σ = ±65 με) [36], which is larger than the AQ8603’s nominal precision of ±50 με. The measurement error of the NBX-6000 (produced by Neubrex Co. Ltd., Kobe, Japan) based on pulse-prepump Brillouin Optical Time Domain Analysis (PPP-BOTDA) was about ±80 με (2σ, σ = ±40 με), which is larger than the NBX-6000’s nominal precision of ±25 με [37]. The second type of FBG sensor comprises a grating packaged with a metal capillary [38,39]. Then the capillary-encapsulated FBG sensors are bonded on the surface of the strand by epoxy resin to measure the elongation. The shortcomings of this type of FBG sensors also exist. First, the elastic modulus of steel is far greater than that of resin, and the mechanical strength of most epoxy resins is limited. Thus, mechanical creep may occur in epoxy, when the epoxy is stressed to beyond 50% of its tensile strength [40], and the bonding may fail if the transmitting stress reaches 80% of the ultimate tensile strength of resin [41]. Second, the sensor is proposed to install in the space between two adjacent steel wires [39]. This installation method may lead to accidental damage to the sensor caused by the dislocation of adjacent wires during strand tensioning. Finally, because the strain distributed in each wire is not identical, the measurements from the sensor for a single wire may have a remarkable difference from the true strain of the strand.

We propose that the long-gauge fiber Bragg grating (LFBG) sensor [42] could overcome the abovementioned limitations. The LFBG sensor has a sensing gauge ranging from 0.1 to 1 m suggesting that the measured strain can represent the average elongation of all wires in the gauge length. Meanwhile, the LFBG sensor is packaged by epoxy-soaked fiber reinforced polymer (FRP) material because the elastic modulus of the epoxy-soaked FRP material is less than that of steel so that the bonding can safely transmit the strain from the strand to the sensor. Moreover, the durability of the LFBG sensor has been verified by the fatigue experiment and long-term durability tests in acidic, alkaline, and salt environments [43]. The applicability of the LFBG sensor was also confirmed in practical monitoring for measuring dynamic strain [44] and dynamic displacement [45], and observing the change in the neutral axis position [46]. However, little is known on the monitoring and calculation of prestress losses based on the LFBG sensor.

This paper is organized as follows: Section 2 introduces the structure and design of an LFBG sensor for prestress monitoring in a strand and proposes the installation procedure in practical operation. Based on the strain measurements from the installed LFBG sensors, Section 3 proposes a method to calculate the itemized prestress losses in both pre-tensioning and post-tensioning. Section 4 and Section 5 demonstrate the application of the proposed method experimentally and via in-site monitoring, respectively.

## 2. The Design and Installation of LFBG Sensors

### 2.1. Introduction of the LFBG Strain Sensor

The structure of an LFBG sensor proposed by Li [42] is illustrated in Figure 1. A notable feature of this sensor is the use of an embedded and hollow polytetrafluoroethylene tube, inside which an FBG is sleeved and fixed at both ends, and the gauge length of the sensing part can be predetermined.

Moreover, the specific design is advantageous for numerous reasons: (i) The hollow tube used to encapsulate the FBG inside can ensure the strain at each point of the fiber optic is identical, and the measurement from the FBG equals the average strain over the gauge length directly. (ii) A special epoxy resin used to recoat the FBG can effectively prevent the slippage between the bare fiber optic and the epoxy resin. Meanwhile, the strain compatibility can be achieved between the FBG and epoxy resin until the measurement attains the breaking strain. This point is important for high stress-strain measurement in practical prestress monitoring. (iii) The bonding capability of the FRP material with structural materials, such as steel and concrete, is excellent. The FRP also has an excellent long-term durability and stability to prevent degradation due to corrosion and extreme environments.

### 2.2. Length Design of LFBG Sensor Installed on the Strand

As shown in Figure 2, the strand is composed of a linear core wire and six helical wires. Because the gauge lengths are no longer than several centimeters, most traditional strain sensors, such as the electric resistance strain gauge and the short-gauge FBG sensor, can obtain only the strain of one wire in the strand. However, this measurement cannot represent the true elongation of the strand. Therefore, the sensing part of the LFBG sensor needs to have a length that can touch all the six helical wires in a spiral. For example, the gauge length of an LFBG sensor is about 20–25 cm for stress monitoring to a 7-wire strand. Thus, the entire length of the LFBG sensor can reach 30–35 cm considering that bonding length of each end is set to about 5 cm. Generally, the total thickness of the sensor and the surrounding epoxy resin is less than 3 mm; this thickness is remarkably less than the diameter of a wire and space between the strand and duct.

### 2.3. Installation Procedure of the LFBG Sensor

The installation procedure is designed to prevent the installed LFBG sensors from accidental failure caused by friction between adjacent strands and between the strand and duct. The restraining block is used to separate adjacent strands and provide space to the sensor. Figure 3a,b show the designed restraining blocks for a 7-strand tendon and 3-strand tendon, respectively. Every block is assembled of two symmetrical parts by connecting bolts. The central hole of the block can contain the core strand and six helical strands that are separated by six grooves. The practical sketch of the restraining blocks on a 7-strand tendon is illustrated in Figure 3c. The distance over 50–60 cm between the two restraining blocks is long enough to contain the LFBG sensors installed on the strands.

As shown in Figure 4a–f, the installation procedure is outlined as follows:(1)Mark the corresponding region on the corrugated pipe. Then let the strands pass through the marked corrugated pipe.(2)Peel the marked region of the corrugated pipe to expose the inner tendon. Clean the surface of the exposed tendons.(3)Install the restraining blocks and tighten the bolts.(4)Attach the LFBG sensors on the surface of the strands. The attachment position of the sensor on each outer strand should be pointed at and close to the core strand.(5)Let the optical cable pass through a protective sleeve and connect to the sensors.(6)Connect the protective sleeve to the corrugated pipe and use epoxy resin to seal off the contact area. Then the protection sleeve inside which the optical cable is placed can be extended away from the corrugated pipe to the nearest vent hole or drain hole.

## 3. The Calculation Method for Itemized Prestress Losses Based on the LFBG Measurements

The discussion proposes an optimized sensor configuration for pre-tensioning and post-tensioning and gives the calculation method for itemized prestress losses in both pre-tensioning and post-tensioning based on the LFBG measurements.

### 3.1. The Itemized Prestress Losses

In the Chinese Code [47], the total prestress loss comprises seven itemized prestress losses named as *σ*_l1_–*σ*_l7_. *σ*_l1_ is the anchorage-seating loss. *σ*_l2_ is the frictional loss containing the loss due to the friction between tendons and duct (*σ*_l2,I_) and the loss due to draw-in of the wedge (*σ*_l2,II_). *σ*_l3_ is the loss due to the temperature difference between the tendon and the abutments in concrete curing. *σ*_l4_ is the loss due to steel relaxation. *σ*_l5_ is the loss due to creep and shrinkage of concrete. *σ*_l6_ is the loss due to the case in which spiral prestressed rebar in annular structure, such as nuclear containment vessel, is extruded by adjacent concrete. *σ*_l7_ is the loss due to elastic shortening of concrete.

These seven itemized prestress losses are broadly classified into two groups: (1) immediate reductions during prestressing of the tendons and the prestress transferring from the tendons to the concrete members; and (2) time-dependent reductions occurring gradually during the in-service life of the structures. The immediate reductions contain *σ*_l1_, *σ*_l2_, *σ*_l3_, *σ*_l6_, and *σ*_l7_. *σ*_l4_, and *σ*_l5_ belong to the time-dependent reductions.

### 3.2. The Case of Pre-Tensioning

As shown in Figure 5a–c, two abutments are fixed securely at both ends of a prestressing bed, and a high-strength steel tendon is pulled between the abutments before the concrete casting. When the concrete attains the required strength for prestressing, the tendon is cut from the abutments, and the prestress is transferred from the tendon to the concrete member through the bond between them. According to the Chinese Code [47], the total loss *σ* is the sum of several itemized prestress losses shown in Equation (1):(1)$$\sigma =\sigma_{l1}+\sigma_{l2}+\sigma_{l3}+\sigma_{l4}+\sigma_{l5}+\sigma_{l7}\;$$
and the term *σ*_l2,I_ does not exist in pre-tensioning.

A pre-tensioning beam is a typical kind of simply-supported beam. As illustrated in Figure 5, the LFBG sensors can be set on these regions of the tendon as follows: (1) regions near the ends of the beam (R1 and R3); (2) region near the mid-span of the beam (R2). The reason for the former choice is that *σ*_l1_ and *σ*_l2,II_ constitute the main part of immediate losses, and *σ*_l3_ and *σ*_l4_ can be considered to be uniformly distributed along the tendon. The reason for the latter is that the mid-span usually has the maximum moment under the action of daily loads.

Five stages (Stages a–e) exist in a prestressed structure from pre-tensioning to load bearing. Of those, Stages a–d are shown in Figure 5a–c, and Stage e represents the in-service stage of the structure. The measured strains at R1–R3 at Stages a–d are set as $\varepsilon_{a}^{Ri}$~$\varepsilon_{d}^{Ri}$ (*i* = 1–3), respectively. The superscript R*i* denotes the variable located at R*i* (*i* = 1–3). The subscript a-d implies that the variable is used at Stages a-d, respectively.

At the tensioning stage (Stage a), the relationship between the tensioning force and monitored strain is:(2)$$F=A_{t}E_{t}\varepsilon_{a}^{Ri}\;(i=\;1–3)$$
where *A_t_* and *E_t_* are the area and elastic modulus of prestressing tendon, respectively. *F* is the tensioning force obtained by the load cell. The value of *E_t_* is determined by a tensile test carried out in the laboratory. No prestress loss occurs at this stage.

*σ*_l1_ and *σ*_l2_ occur at Stage b of transferring the tensioning force from jack to prestressing bed. However, it is difficult to divide them without using special measurements, so the sum of *σ*_l1_ and *σ*_l2_ is shown as follows:(3)$$\sigma_{l1}+\sigma_{l2}=\sigma_{l1}+\sigma_{l2,\mathrm{II}}\approx E_{t}(\varepsilon_{a}^{R1}-\varepsilon_{b}^{R1})\;$$

At Stage c of concrete member curing, *σ*_l3_ and $\sigma_{l4}^{Ri}$ begin to appear. Because the low-relaxation prestressing strand can finish its relaxation in several hundreds of hours, $\sigma_{l4}^{Ri}$ can be obtained entirely in Stage c. Thus, *σ*_l3_ and $\sigma_{l4}^{Ri}$ can be respectively calculated by Equations (4) and (5):(4)$$\sigma_{l3}=E_{t}\alpha_{t}\cdot \Delta t\;$$
(5)$$\sigma_{l4}^{Ri}=E_{t}(\varepsilon_{b}^{Ri}-\varepsilon_{c}^{Ri}-\alpha_{t}\cdot \Delta t)\;(i=\;1\sim 3)$$

At Stage d, the tendons between the beam and the abutment are cut off, and the prestress is resisted by the entire section of the beam. At this time, $\sigma_{l7}^{Ri}$, which can be obtained by Equation (6) comes into play:(6)$$\sigma_{l7}^{Ri}=E_{t}(\varepsilon_{c}^{Ri}-\varepsilon_{d}^{Ri})\;(i\;=\;1\sim 3)$$

At the in-service stage (Stage e), $\sigma_{l5}^{Ri}$ can be obtained by Equation (7) when the live load is not applied on the structure:(7)$$\sigma_{l5}^{Ri}=E_{t}(\varepsilon_{d}^{Ri}-\varepsilon_{e}^{Ri})\;(i\;=\;1\sim 3)$$

Finally, substituting Equations (3)–(7) into Equation (1), the total prestress losses at different locations of the structure are obtained.

### 3.3. The Case of Post-Tensioning

A remarkable limitation of the pre-tensioning system is that the tendons always have to be straight. However, the post-tensioning system enables the tendons to keep a curved profile before and after tensioning. The ducts inside which the tendons are placed can be fixed to the reinforcements to remain in the desired profile. Then, once the concrete reaches the desired strength, the tendons are tensioned and anchored using external anchors rather than depending on the bond between tendon and concrete as in the pre-tensioning case. Figure 6a–b show the two stages of the post-tensioning procedure.

According to the Chinese Code [47], the total loss *σ* is the sum of several itemized prestress losses shown in Equation (8):(8)$$\sigma =\sigma_{l1}+\sigma_{l2}+\sigma_{l4}+\sigma_{l5}+\sigma_{l6}+\sigma_{l7}\;$$

As shown in Figure 6, *σ*_l6_ is zero. *σ*_l2,I_ exists and does not equal to zero because the profile of the duct is usually curved. *σ*_l7_ is zero unless the tendons are tensioned batch-wise.

The LFBG sensors can be set on the three regions (R1–R3) shown in Figure 6. In post-tensioning, the curve of the duct profile can be described as a combination of three parabolas. The two linking points between the three parabolas are key points in the design of prestressed structures. Thus, R1 and R3 are set to be near to these two linking points. R2 is set at the mid-span of the beam.

There are three stages (Stages a~c) in a post-tensioned structure from post-tensioning to load bearing. Stages a and b are the tensioning stage and the stage of anchoring, respectively. Stage c is the stage of grout curing and in-service stage of the structure that is shown in Figure 6b.

At Stage a, there are only the frictional losses *σ*_l2_. In Equation (9), $\sigma_{l2}^{Ri}$ is the frictional loss in R*i*:(9)$$\sigma_{l2}^{Ri}=F/A_{t}-E_{t}\varepsilon_{a}^{Ri}\;(i=\;1\sim 3)$$

At Stage b, the stress variation equals $\sigma_{l1}^{Ri}$ as follows:(10)$$\sigma_{l1}^{Ri}=E_{t}\left(\varepsilon_{a}^{Ri}-\varepsilon_{b}^{Ri}\right)\;$$

At Stage c, the sum of $\sigma_{l4}^{Ri}$ and $\sigma_{l5}^{Ri}$ is shown in Equation (11). It also needs to be measured without live load action:(11)$$\sigma_{l4}^{Ri}+\sigma_{l5}^{Ri}=E_{t}\left(\varepsilon_{b}^{Ri}-\varepsilon_{c}^{Ri}\right)\;$$

This proposed method is also suitable for prestress loss calculation of more than one tendon in pre-tensioning and post-tensioning. Moreover, it is necessary to keep a distance between the sensor-placed regions near both ends of the strand and the anchor, since the violent variation of stress may threaten the safety of the LFBG sensors in prestress releasing. Finally, all the strain measurements should be updated by temperature compensation.

## 4. Verification for The Prestress Loss Monitoring Using LFBG Sensor: Experiment

This experiment in this study has two main purposes: to verify the applicability of the LFBG sensor to measure the prestress of a tendon and to investigate the accuracy of the proposed calculation method of prestress loss. In order to correspond to the proposed prestress loss monitoring methods in Section 3, this experiment includes two parts: pre-tensioning test and post-tensioning test.

### 4.1. Pre-Tensioning Test

#### 4.1.1. Test Design

As shown in Figure 7a, a 7-wire strand inserted into a hollow steel tube was fixed at both abutments of a prestressing bed. The tube was held by two supports and separated from the strand. The strand was placed at the center of the tube. The lengths of the strand and hollow steel tube were 3000 mm and 2500 mm, respectively. The nominal diameter and elastic modulus of the strand were 15.2 mm and 200 GPa, respectively. The outside and inside diameters of the hollow tube were 50 mm and 48 mm, respectively. Three monitored regions (R1–R3) with a uniform length of 250 mm were set from the right end to the mid-point of the strand, and the distance between adjacent regions was 250 mm. Three LFBG sensors (S1–S3) with a uniform gauge length of 250 mm were placed on R1–R3, respectively. In each region, each helical wire was attached to a foil strain-gauge (FSG) to measure the strain precisely. The numbering rule is as follows. For example, the 6 FSGs in R1 are named as E11–E16. The first number means the sensor is in R1 and the second number represents the number of the wire. Details are given in Figure 7c about the sensor placement on the wire.

The process of loading can be divided into three steps (Step I–III). At Step I, the increasing load *F* was applied by a jack to tension the strand through eight successive loading steps from 0 to 156 kN with an increment of 20 kN. After the final loading step, tensioning reduces from 156 kN to 149.8 kN because of the anchorage-seating loss and the loss due to draw-in of the wedge. At the beginning of Step II, the tube was filled with grout. When the strength of the grout exceeded 50 MPa (48 h after tensioning), the supports were removed. At Step III, a vertical load *P* was divided equivalently into two parts by a transferring steel board and applied at two points 500 mm away from both ends of the tube. *P* was increased with a loading step of 3 kN from 0 kN to 24 kN. Details about Step II and Step III are respectively shown in Figure 7b,d,e. The strains of R1–R3 in each helical wire at Steps I-III were measured by FSGs and LFBG sensors, respectively. In addition, the temperature was kept constant during the entire experiment to avoid the expansion or contraction of the abutments and strand. All measured data were updated by temperature compensation.

#### 4.1.2. Results and Analysis

Table 1 gives the measured strains from the FSGs, the average values of the measured strains from the FSGs, and the measured strains from the LFBG sensors at Step I. Table 2 shows the comparison between the true stresses, the stresses calculated from the average values of the measured strains from the FSGs, and the stresses calculated from the strains obtained from the LFBG sensors. Four remarkable features are notable: First, most calculated stresses from sensor measurements are lower than the true stresses, the result ascribed to the gap between the adjacent wires. Second, the difference between the maximum and minimum strains of the six helical wires in the same region can approach or exceed 10% of the applied strain.

This result implies that the strain measured in one wire only does not necessarily represent the elongation of the whole strand. Moreover, the measured strains from the LFBG sensors are between the maximum and minimum strains from the FSGs and close to the average values of the measured strains from the FSGs. This result is attributed to the fact that LFBG can acquire the average elongation of six helical wires because the sensing part of the LFBG can cover the six wires.

Finally, most of the differences between the true stresses in the strand and the calculated stresses by strains from LFBG sensors are less than 5%. The maximum error is 6.0%. However, the differences between the true stresses in the strand and the calculated stresses from the average values of the measured strains from the FSGs are over 5%. The maximum error is up to 9.2%. It appears that the errors in strains from LFBG sensors are approximately half of those obtained from FSGs. An important reason behind this effect is that the FSGs attached on the helical wires are not parallel with the tensioning direction. In summary, these features show LFBG sensor is better suited to monitor prestress in the strand than common FSG.

Table 3 gives the monitored strains from different sensors at Step II. The comparison between the true stresses and the calculated stresses are shown in Table 4. Most of the losses calculated from the strains obtained from the LFBG sensors are approximate to or less than the corresponding losses calculated from the average values of the measured strains from the FSGs. Moreover, some FGSs become invalid at the end of this step. It further shows that the traditional electrical sensor may not satisfy the requirements of “local measurement in duct” due to the lack of long-term durability.

Before the beginning of Step III, the values of *P*, *F*, and sensors are set to zero again based on the measured data in Step II. The measured strains at each loading step of Step III are shown in Table 5. Note that the relationship between *F* and strand strains does not match Equation (1) anymore because the force is undertaken by not only strand but also solid grout and steel tube. There are two notable characteristics in Table 5. On the one hand, the number of damaged FSGs grows with increasing load. By contrast, all LFBG sensors can measure the strains well in the entire loading process. On the other hand, although the strains in R1–R3 should be close in theory, the difference between the measured strains from S1–S3 is less than 10% only in the case of *P* ≤ 12 kN. This phenomenon may be attributed to the crack in grout occurring near R1 and R2. The chief reason for that is when *P* = 12 kN, the measured strain is about 150 με, which is close to the threshold of the tensile strain of most concrete. This phenomenon shows the durability of LFBG sensors in the case of grout cracking in practical prestress monitoring.

Based on Equations (3)–(7), Table 6 lists the itemized prestress losses calculated from the monitored strains. Because the temperature was kept constant during the test, *σ*_l3_ is zero. In addition, *σ*_l7_ is zero because the strand was not cut. *σ*_l5_ is also zero because the interval between grout curing and loading was short. Compared with the losses calculated from the average values of the measured strains from the FSGs, the losses calculated from the measured strains from the LFBG sensors were closer to the true losses. The error in the latter is only half of that in the former. This result verifies that LFBG sensor has more accuracy than traditional FSG in prestress monitoring.

### 4.2. Post-Tensioning Test

#### 4.2.1. Test Design

Details of the dimensions and reinforcement configuration of a simply-supported beam used in the experiment are shown in Figure 8a. The total length of the beam was 6000 mm, with a span of 5400 mm. The cross-section had a rectangular shape with 220 mm width and 450 mm depth. The compressive strength of the concrete was about 39 N/mm^2^. The elastic modulus and Poisson’s ratio of the concrete were 3.03 × 10^4^ N/mm^2^ and 0.19, respectively. A curved duct with a diameter of 50 mm was embedded into the beam. The process of the test can be divided into two steps (Step I–II). At Step I, a 3-strand tendon was passed through the duct and tensioned by the jack with an increasing load from 0 to 120, 240, 360, 480 and 540 kN. To counteract the frictional loss, we overloaded the final tension force to 555 kN that is 3% higher than 540 kN. Because of the effect of the anchorage-seating loss *σ*_l1_, the tensioning force reduced from 555 kN to 426.6 kN in the case of anchoring. At Step II, it took 72 h to observe the steel relaxation and creep of the concrete and for the strength of grout to reach 50 MPa.

As shown in Figure 8a, we selected five regions located at the right linking point, 1/3 span, mid-span, 2/3 span, and left linking point (R1–R5) of the top strand for monitoring. Five LFBG sensors with a uniform gauge length of 250 mm were placed on R1–R5. 30 FSGs were successively attached on R1–R5 of the six helical wires of the strand. The numbering mode for the LFBG sensors and FSGs was the same as that in the last test. More details about the sensor installation and loading are illustrated in Figure 8b.

#### 4.2.2. Results and Analysis

Different from the pre-tensioning case, the true stresses on R1–R5 cannot be obtained due to the frictional prestress loss distribution. Therefore, the theoretical prestress loss predictions by the Chinese Code [47] are used to replace the true prestress losses in the comparisons of *σ*_l2_ and *σ*_l1_. Table 7 gives the monitored strains from the FSGs and LFBG sensors at Step I. The differences between the maximum and minimum strains obtained from different wires also reaches or exceeds 10% of the applied strain. Assuming the average strain is approximate to the true strain, the large dispersion in the strain measurements demonstrates that the traditional short-gauge strain sensor is not suitable in prestress monitoring. Because the extending direction of the LFBG sensors is parallel to the tensioning force, most of the strains from LFBG sensor are larger than the average strains from FSG measurements at each loading step. This phenomenon also occurs in the previous test of pre-tensioning. Using Equation (9), Table 8 shows *σ*_l2_ and *σ*_l1_ calculated from LFBG sensor measurements at each loading step. Figure 9a,b show the comparisons between *σ*_l2_ and *σ*_l1_ calculated from the measured strains from the LFBG sensors and the theoretical predictions of prestress losses by the Chinese Code [47]. The developing trend of the calculated *σ*_l2_ and *σ*_l1_ are close to the loss profiles of theoretical predictions although there are some deviations between the calculated values and the predictions. This result also proves that the calculations for *σ*_l2_ and *σ*_l1_ are correct based on the strain measurements from the LFBG sensors.

At Step II, Table 9 lists the strain decrements of R1–R5 in 72 h due to (1) steel relaxation of the strands; (2) creep and shrinkage of the concrete. The sum of the corresponding prestress losses, *σ*_l4_ and *σ*_l5_, are listed in Table 10. Compared with the data in Table 4, the losses in Table 10 imply that the major parts of the losses are triggered by the creep and shrinkage of concrete. Also, several FSGs are observed to be invalid in this step owing to the immersion of grout. The fact that the FBG sensors can measure during 72 h of grout curing again shows the durability to the sensing part of the LFBG sensor.

## 5. Verification for the Prestress Loss Monitoring Using LFBG: In-Site Monitoring

The specific aims regarding in-site monitoring are to observe whether the LFBG sensors installed by the proposed installation procedure in Section 2.3 are valid in long-term monitoring, and to obtain the itemized prestress losses in practical post-tensioning to ensure the security of the construction.

### 5.1. Member Fabrication and Sensor Placement

This practical prestress loss monitoring was based on the project of Multifunctional Drama Hall of the Fuzhou Straits Cultural Art Center in Fujian Province, China. The monitored beam had a length of 18.475 m. The width and the height of the beam were 1.3 m and 0.7 m, respectively. Details of the dimensions and reinforcement configuration are shown in Figure 10. Three 7-strand tendons were passed the curved duct, and all strands in three tendons had an identical nominal diameter of 15.2 mm. The ultimate tensile strength and elastic modulus of the strand were 1860 MPa and 200 GPa, respectively. The tensioning was controlled by the oil-pressure gauge of the jack used to apply tensioning to the tendons. There are three loading steps including 10 MPa, 20 MPa, and 28 MPa shown in the oil-pressure gauge to apply the prestresses of 358.40 MPa, 728.0 MPa, and 1023.7 MPa to the tendon, respectively. The applied prestress in the tendon at the final loading step contained an overstressing of 3% to counteract the frictional loss. Due to the anchorage-seating loss *σ*_l1_, the applied prestress decreased from 1023.7 MPa to 862.3 MPa.

The six outer strands of the mid-tendon named as w1–w6 were used for prestress monitoring. As shown in Figure 10, two monitored regions were set nearby the linking points in the strands. 12 LFBG sensors were fixed in the regions. The numbering mode for the LFBG sensors is as follows. The six LFBG sensors on R1 of W1–W6 are named as S11–S16. The first number means the sensor is in R1, and the second number represents the number of the strand. The other six LFBG sensors on R2 of W1–W6 are named as S21–S26. At tensioning, the sensors measured the strain increments of each strand in each loading step. When the tensioning was finished, and the prestress began to transfer to the beam, the strain data were measured continuously. The entire monitoring lasted about 90 days, and all data were updated by temperature compensation.

Moreover, all LFBG sensors were installed by the installation method proposed in Section 2.3. Figure 11a–d show the whole process of sensor placement.

### 5.2. Results and Analysis

Table 11 lists the measured strains of the six helical strands and the average strains at each loading step in tensioning. The strains distributing in the strands are quite dispersed, and the deviation between the maximum and minimum strain can reach 50% of the average applied strains. This phenomenon implies that some strands such as w2 and w3 were still loose before the tensioning was applied. Moreover, the strand w1 may be ruptured earlier than other strands because it undertakes the extra prestress. Thus, a remarkable conclusion that can be derived from the data shown in Table 11 is that monitoring prestress in a tendon merely from the measured strains in one of the six helical wires using the traditional “short-gauge” strain sensor may ignore the uneven strain distribution in the same section. Table 12 and Table 13 give *σ*_l2_ and *σ*_l1_ calculated from the measured strains and their prediction based on the Chinese Code [47]. The fact that the calculated losses are close to the predictions implies that the prestress loss calculations are correct in principle based on the strain measurements from the LFBG sensors.

Table 13 records the strain reductions of W1–W6 due to steel relaxation of strands and creep of concrete in 90 days after anchoring. The calculated sum of the prestress losses, *σ*_l4_ and *σ*_l5_, is listed in Table 14 and illustrated in Figure 12.

There are three features that may be observed in Table 14 and Figure 12. First, the phenomenon that all LFBG sensors can trace the stress variation proves that the LFBG has enough durability for long-term prestress loss monitoring, and the proposed installation method is valid. Second, the increment of the sum of *σ*_l4_ and *σ*_l5_ in each strand has a similar trend. It proves the stability of the LFBG in measurements. Finally, almost 50% losses are completed in 48 h and the monitored data become stable after the 51st day. This feature shows that the prestress loss monitoring should be applied at an early stage of prestressing and last until the loss data tends to stabilize.

## 6. Conclusions

We demonstrated the deployment of the LFBG sensor for prestress loss monitoring and Evaluation. An appropriate length design of LFBG sensor and the installation method were proposed. Then we showed the calculation methods for the itemized prestress losses in pre-tensioning and post-tensioning. The applicability of the LFBG for prestress loss monitoring was verified by experiments in the laboratory and in-site monitoring. From the results and discussions, the following conclusions can be drawn:(1)An appropriate gauge length for LFBG sensor is at least 25 cm for prestress loss monitoring in the strand because the gauge can obtain the average strain by covering the six helical wires.(2)Severe frictions between the strand and duct and the grout crack can bring accidental damage to LFBG sensor. The proposed installation method can prevent the LFBG sensor from these ruptures effectively occurring at not only tendon tensioning but also structure loading. The durability and stability of the LFBG sensor are proved to be better than those of traditional FSGs.(3)The proposed calculation method acquired the itemized prestress losses at different stages of applying pretension accurately. Our results from the experiments including the cases of pre-tensioning and post-tensioning showed that the losses calculated from the measured strains of the LFBG sensors were more precise compared to those calculated from traditional FSGs. Moreover, from the in-site monitoring, we obtained the uneven stress distribution in different strands, measured the immediate losses at tensioning, and traced the time-dependent losses for 90 days. Thus, this calculation method can be easy to apply in the itemized prestress losses monitoring.(4)Compared with the traditional electrical sensor, the LFBG sensor is proved to have better durability for long-term prestress loss monitoring in practice, especially in the case of grout cracking and aggressive environment.

## Figures and Tables

**Figure 1 sensors-18-04106-f001:**
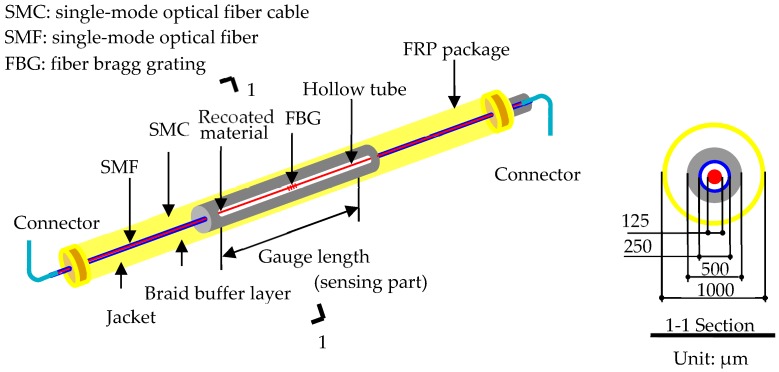
The structural design of the packaged LFBG sensor proposed by Li [42].

**Figure 2 sensors-18-04106-f002:**
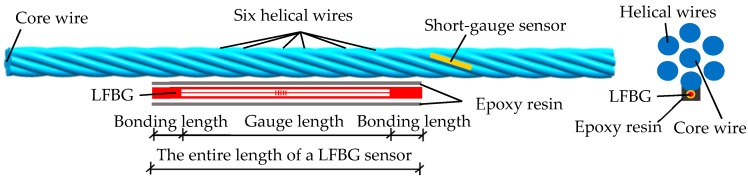
Comparison of gauge length between LFBG sensor and the short-gauge sensor.

**Figure 3 sensors-18-04106-f003:**
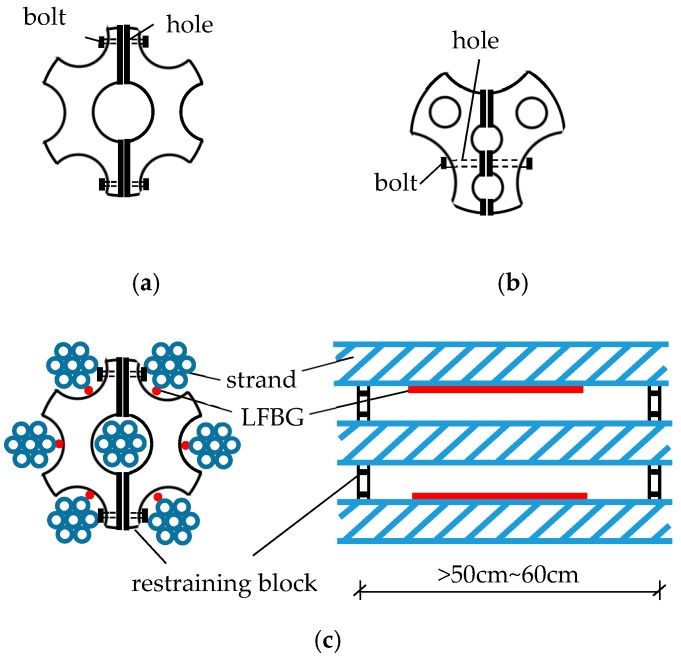
Sketches of the restraining block for (**a**) 7-strand tendon; (**b**) 3-strand tendon; and (**c**) for separating adjacent strands.

**Figure 4 sensors-18-04106-f004:**
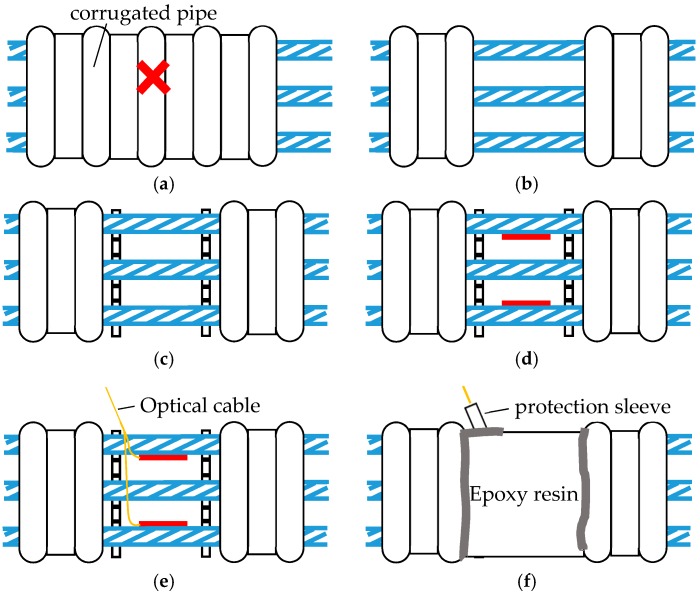
The installation procedure of the LFBG sensor. (**a**) Place the strands through the marked pipe; (**b**) strip the partial corrugated pipe; (**c**) install the restraining blocks; (**d**) attach the LFBG sensors; (**e**) connect the sensor to the optical cable; and (**f**) connect the protection sleeve.

**Figure 5 sensors-18-04106-f005:**
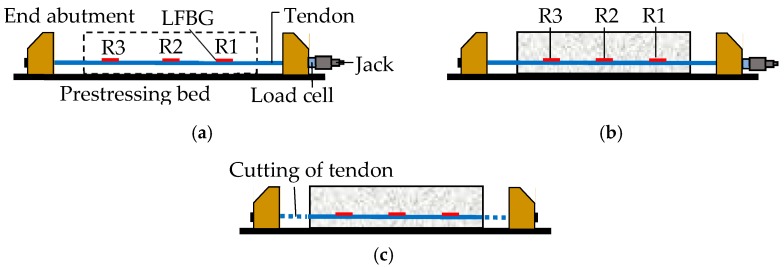
Three stages of pre-tensioning: (**a**) applying prestress to tendons; (**b**) casting and curing of concrete member; and (**c**) cutting of tendon.

**Figure 6 sensors-18-04106-f006:**
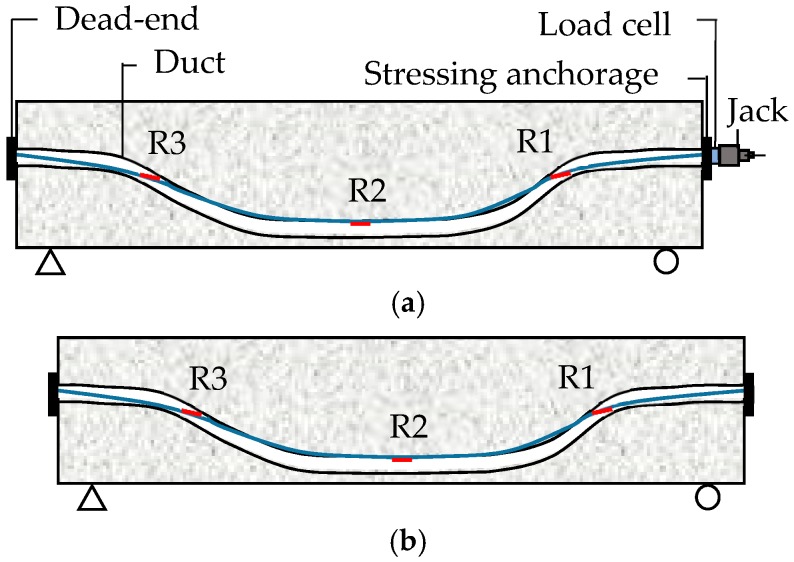
The schematic of the two stages of post-tensioning: (**a**) Application of tensioning to tendons; and (**b**) Fitting the wedge and cutting the tendon.

**Figure 7 sensors-18-04106-f007:**
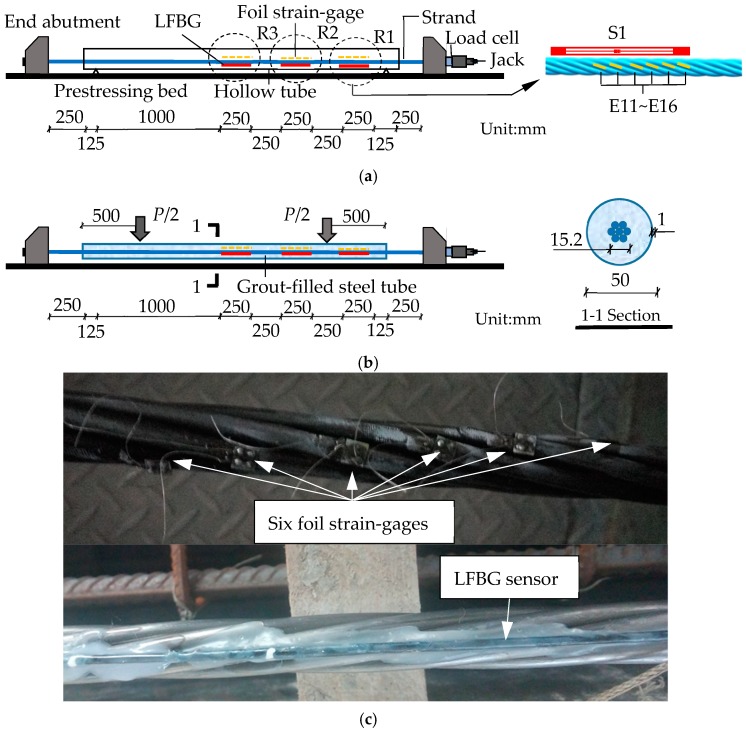
The sketches and photographs of the test design: (**a**) applying tensioning to the bare strand; (**b**) applying load on the cement-filled steel tube; (**c**) the photograph of the strand attached with sensor; (**d**) the photograph of tensioning and (**e**) the photograph of vertical load applied on the cement-filled steel tube.

**Figure 8 sensors-18-04106-f008:**
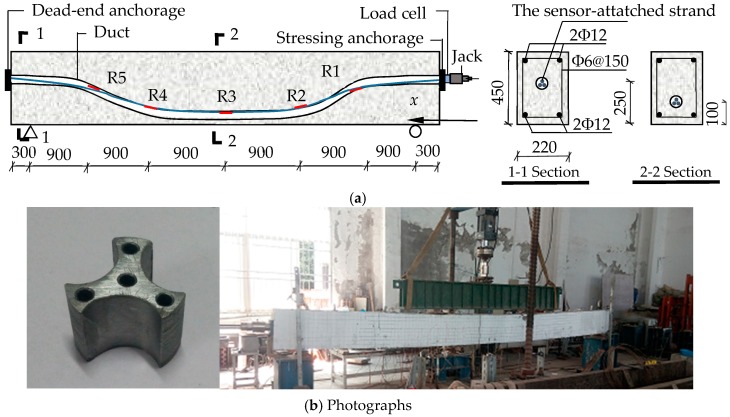
Sketches and photographs of the experiment. (**a**) Applied tensioning to tendons; and (**b**) photographs of the restraining block and tensioning.

**Figure 9 sensors-18-04106-f009:**
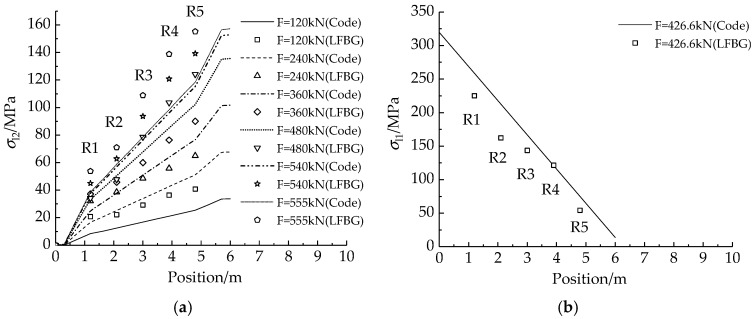
The comparisons of the two calculated losses and the predictions according to the Chinese Code [44]: (**a**) *σ*_l2_; and (**b**) *σ*_l1_.

**Figure 10 sensors-18-04106-f010:**
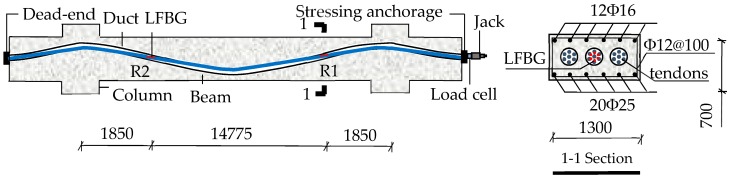
The schematic of the monitored beam in in-site measurement.

**Figure 11 sensors-18-04106-f011:**
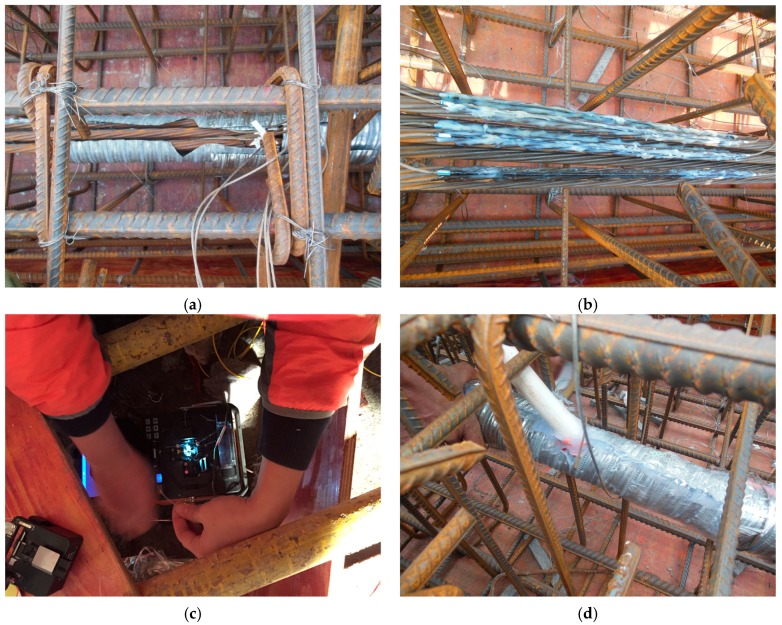
Photographs of the LFBG sensor deployment in in-site monitoring: (**a**) Peeling the marked region to expose the tendon; (**b**) attaching the LFBG sensors on the strands; (**c**) connecting the sensor to the optical cable; and (**d**) connecting the protection sleeve to the pipe.

**Figure 12 sensors-18-04106-f012:**
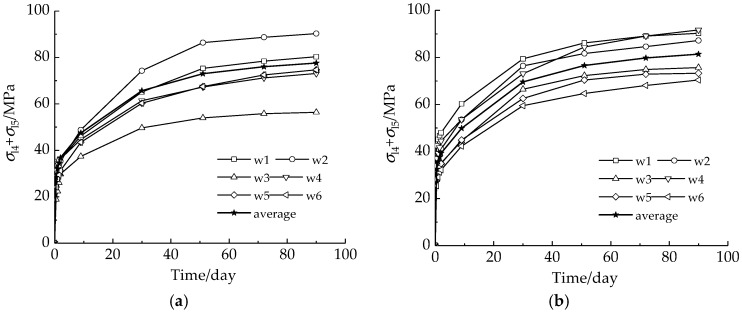
The increment of the sum of *σ*_l4_ and *σ*_l5_ for 90 days: (**a**) R1; and (**b**) R2.

**Table 1 sensors-18-04106-t001:** The monitored strains of the strand at Step I. (Unit: με).

Region	*F/*kN	20	40	60	80	100	120	140	156	149.8
R1	E11	817	1487	2189	2879	3540	4201	4791	5246	5053
	E12	520	1042	1656	2331	3045	3734	4400	5013	4708
	E13	720	1314	1971	2670	3336	3978	4583	5124	4936
	E14	671	1291	1979	2682	3381	4055	4692	5244	5080
	E15	795	1576	2305	3014	3707	4374	5006	5550	5271
	E16	716	1348	2028	2714	3393	4044	4665	5214	5095
	Average strain(FSG) *	**707**	**1343**	**2021**	**2715**	**3400**	**4064**	**4689**	**5232**	**5024**
	S1	**713**	**1426**	**2129**	**2808**	**3524**	**4237**	**4940**	**5482**	**5270**
**R2**	E21	695	1353	2013	2674	3356	3991	4593	5156	4929
	E22	616	1254	1903	2553	3231	3857	4452	5004	4725
	E23	753	1420	2076	2728	3410	4035	4631	5184	4976
	E24	616	1211	1825	2456	3136	3753	4349	4892	4684
	E25	705	1385	2062	2740	3460	4105	4727	5279	5149
	E26	687	1329	1973	2616	3302	3911	4500	5031	4828
	**Average strain(FSG)**	**678**	**1325**	**1975**	**2628**	**3316**	**3942**	**4542**	**5091**	**4882**
	**S2**	**685**	**1388**	**2035**	**2776**	**3425**	**4077**	**4814**	**5238**	**5028**
R3	E31	707	1382	2080	2760	3384	4060	4716	5236	5014
	E32	683	1359	2063	2756	3391	4073	4739	5265	5044
	E33	705	1318	1959	2592	3187	3824	4456	4946	4826
	E34	737	1425	2129	2820	3465	4135	4801	5308	4986
	E35	621	1298	2004	2698	3350	4020	4690	5202	4979
	E36	694	1379	2090	2788	3447	4120	4800	5307	5082
	Average strain(FSG)	**691**	**1360**	**2054**	**2736**	**3371**	**4039**	**4700**	**5211**	**4989**
	S3	**711**	**1407**	**2064**	**2764**	**3500**	**4186**	**4900**	**5356**	**5128**

* Average strain (FSG) represents the average value of the strains from the six FSGs obtained from the same region.

**Table 2 sensors-18-04106-t002:** The measurement errors in calculated stresses from the strains obtained from the FSGs and LFBG sensors.

	*F/*kN	20	40	60	80	100	120	140	156	149.8
	True Stress/MPa	143.0	285.7	428.6	571.4	714.3	857.1	1000.0	1114.3	1070.0
R1	Stress(FSG) */MPa	141.4	268.6	404.2	543.0	680.0	812.8	937.8	1046.4	1004.8
	Error/%	−1.1	−6.0	−5.7	−4.8	−4.6	−6.2	−5.9	−6.1	−6.1
	Stress(LFBG) **/MPa	142.6	285.2	425.8	561.6	704.8	847.4	988.0	1096.4	1054.0
	Error/%	−0.3	−0.2	−0.7	−1.7	−1.3	−1.1	−1.2	−1.6	−1.5
R2	Stress(FSG)/MPa	135.6	265.0	395.0	525.6	663.2	788.4	908.4	1018.2	976.4
	Error/%	−5.0	−7.3	−7.8	−8.0	−7.2	−8.0	−9.2	−8.6	−8.7
	Stress(LFBG)/MPa	137.0	277.6	407.0	555.2	685.0	815.4	962.8	1047.6	1005.6
	Error/%	−4.0	−2.8	−5.0	−2.8	−4.1	−4.9	−3.7	−6.0	−6.0
R3	Stress(FSG)/MPa	138.2	272.0	410.8	547.2	674.2	807.8	940.0	1042.2	997.8
	Error/%	−3.5	−4.8	−4.2	−4.2	−5.6	−5.8	−6.0	−6.5	−6.7
	Stress(LFBG)/MPa	142.2	281.4	412.8	552.8	700	837.2	980.0	1071.2	1025.6
	Error/%	−0.6	−1.5	−3.7	−3.3	−2.3	−1.9	−2.0	−3.9	−4.1

* Stress (FSG) is the stress calculated from the average strain (FSG) in Table 1. ** Stress (LFBG) is the stress calculated from the measured strains by LFBG sensors in Table 1.

**Table 3 sensors-18-04106-t003:** The monitored strains of the strand at Step II. (Unit: με).

	Time/Hour	0	1	2	3	12	24	48
	F/KN	149.8	149.55	149.35	149.24	149.1	148.95	148.9
R1	E11	5053	5046	5040	5038	5032	5015	5009
	E12	4708	4697	4689	4685	4674	4662	4656
	E13	4936	4928	4921	4920	4918	-	-
	E14	5080	5066	5061	5058	5044	5027	5022
	E15	5271	5265	5261	5257	5249	5234	5226
	E16	5095	5087	5081	5077	5068	5052	5044
	Average strain of E11–E16	**5024**	**5015**	**5009**	**5006**	**4998**	**4998**	**4991**
	S1	**5270**	**5260**	**5253**	**5249**	**5242**	**5238**	**5234**
R2	E21	4929	4918	4909	4904	4893	4880	4865
	E22	4725	4720	4717	4715	4709		-
	E23	4976	4967	4960	4958	4951	4931	4909
	E24	4684	4673	4666	4662	4657	4633	4624
	E25	5149	5143	5138	5135	5130	5113	5094
	E26	4828	4820	4814	4810	4801	4777	4764
	Average strain of E21–E26	**4882**	**4874**	**4867**	**4864**	**4857**	**4867**	**4851**
	S2	**5028**	**5019**	**5013**	**5009**	**5003**	**4997**	**4995**
R3	E31	5014	5005	4997	4993	4989	4983	4980
	E32	5044	5033	5025	5018	5008	5005	5002
	E33	4826	4817	4810	4808	4801	4799	4795
	E34	4986	4978	4972	4970	4967	4957	4951
	E35	4979	4971	4964	4961	4957	4952	-
	E36	5082	5070	5063	5058	5054	5048	5044
	Average strain of E31–E36	**4989**	**4979**	**4972**	**4968**	**4963**	**4957**	**4954**
	S3	**5128**	**5117**	**5112**	**5108**	**5103**	**5097**	**5094**

**Table 4 sensors-18-04106-t004:** The stresses and the measurement errors calculated from the strains obtained from the FSGs and LFBG sensors.

	Time/Hour	1	2	3	12	24	48
	True stress/MPa	1.78	3.2	4.0	5.0	6.07	6.43
R1	Stress(FSG)/MPa	1.8	3	3.6	5.2	5.2	6.6
	Error/%	1.1	−6.3	−10.0	4.0	4.0	2.6
	Stress(LFBG)/MPa	2	3.4	4.2	5.6	6.4	7.2
	Error/%	12.4	6.3	5.0	12.0	5.4	12.0
R2	Stress(FSG)/MPa	1.6	3	3.6	5	3	6.2
	Error/%	−10.1	−6.3	−10.0	0	50.5	−3.6
	Stress(LFBG)/MPa	1.8	3	3.8	5	6.2	6.6
	Error/%	1.1	−6.3	−5.0	0	2.1	2.6
R3	Stress(FSG)/MPa	2	3.4	4.2	5.2	6.4	7
	Error/%	12.4	6.3	5.0	4.0	5.4	8.9
	Stress(LFBG)/MPa	2.2	3.2	4	5	6.2	6.8
	Error/%	23.6	0	0	0	2.1	5.8

**Table 5 sensors-18-04106-t005:** The monitored strains of the strand at Step III. (Unit: με).

	*P/*kN	3	6	9	12	15	18	21	24
	*F/*kN	0.6	2.2	5.8	10.1	15.1	20.8	26.7	32.7
R1	E11	14	46	89	145	230	318	-	-
	E12	16	63	105	155	258	376	516	682
	E13	-	-	-	-	-	-	-	-
	E14	6	34	73	123	-	-	-	-
	E15	14	48	103	-	-	-	-	-
	E16	8	42	88	153	233	-	-	-
	Average strain(FSG)	**12**	**47**	**92**	**144**	**240**	**347**	**516**	**682**
	S1	**12**	**47**	**97**	**155**	**252**	**367**	**508**	**662**
R2	E21	6	32	75	130	195	279	379	482
	E22	-	-	-	-	-	-	-	-
	E23	11	49	95	150	231	315	-	-
	E24	11	43	92	141	221	312	425	551
	E25	19	59	118	172	-	-	-	-
	E26	22	66	116	174	260	365	-	-
	Average strain(FSG)	**14**	**50**	**99**	**153**	**227**	**318**	**402**	**517**
	S2	**11**	**47**	**93**	**149**	**234**	**335**	**456**	**578**
R3	E31	8	44	91	141	206	275	373	480
	E32	14	50	98	157	236	-	-	-
	E33	9	42	81	122	-	-	-	-
	E34	19	61	-	-	-	-	-	-
	E35	-	-	-	-	-	-	-	-
	E36	6	33	74	124	196	273	384	-
	Average strain(FSG)	**11**	**46**	**86**	**136**	**213**	**274**	**379**	**480**
	S3	**12**	**42**	**89**	**141**	**211**	**297**	**396**	**500**

**Table 6 sensors-18-04106-t006:** Comparison of different prestress losses in the pre-tensioning experiment.

Itemized Prestress Loss		*σ*_l1_ + *σ*_l2, II_	*σ* _l2, I_	*σ* _l3_	*σ* _l4_	*σ* _l5_	*σ* _l6_	*σ* _l7_	Total Loss
True loss/MPa		44.3	- *	0	6.4	0 **	-	0	50.7
Loss(FSG)	Value/MPa	41.6	-	0	6.6	0	-	0	48.2
	Error/%	−6.1	-	0	3.1	0	-	0	−4.9
Loss(LFBG)	Value/MPa	42.4	-	0	6.9	0	-	0	49.3
	Error/%	−4.3	-	0	7.8	0	-	0	−2.8

* “-” denotes this loss does not exist in pre-tensioning case. ** “0” denotes that *σ*_l5_ is small because the interval between grout curing and loading is short.

**Table 7 sensors-18-04106-t007:** The monitored strains of the strand at Step I. (Unit: με).

	*F/*kN	120	240	360	480	540	555	426.6
R1	E11	1371	2775	4193	5682	6368	6471	5214
	E12	1336	2704	4111	5532	6218	6343	5111
	E13	1300	2625	3993	5368	6046	6171	4807
	E14	1257	2543	3786	5250	5846	5954	4968
	E15	1325	2689	4161	5504	6243	6368	5125
	E16	1382	2800	4229	5686	6371	6494	5227
	Average strain(FSG)	**1329**	**2689**	**4079**	**5504**	**6182**	**6300**	**5075**
	S1	**1321**	**2689**	**4089**	**5529**	**6200**	**6336**	**5211**
R2	E21	1357	2729	4179	5589	6289	6421	5546
	E22	1318	2657	4079	5471	6132	6264	5439
	E23	1264	2554	3896	5250	5932	6075	5271
	E24	1236	2504	3807	5154	5825	5936	5154
	E25	1289	2606	4006	5392	6079	6232	5451
	E26	1361	2721	4204	5500	6236	6396	5496
	Average strain(FSG)	**1304**	**2629**	**4029**	**5393**	**6082**	**6221**	**5393**
	S2	**1314**	**2657**	**4046**	**5461**	**6111**	**6250**	**5439**
R3	E31	1321	2693	4125	5461	6150	6261	5461
	E32	1296	2639	4046	5354	6036	6146	5375
	E33	1254	2579	3893	5200	5896	6000	5246
	E34	1229	2475	3829	5161	5686	5871	5164
	E35	1243	2575	3950	5236	5875	5979	5229
	E36	1351	2707	4115	5456	6142	6261	5432
	Average strain(FSG)	**1282**	**2611**	**3993**	**5311**	**5964**	**6086**	**5318**
	S3	**1279**	**2607**	**3975**	**5307**	**5957**	**6061**	**5343**
R4	E41	1282	2646	4025	5321	5986	6082	5404
	E42	1279	2629	4036	5318	6007	6107	5421
	E43	1204	2536	3793	5039	5646	5796	5129
	E44	1189	2414	3743	5036	5579	5654	5079
	E45	1218	2568	3871	5161	5807	5889	5275
	E46	1329	2721	4057	5432	6114	6215	5575
	Average strain(FSG)	**1250**	**2586**	**3921**	**5218**	**5857**	**5957**	**5314**
	S4	**1243**	**2571**	**3893**	**5182**	**5821**	**5911**	**5304**
R5	E51	1236	2389	3861	5246	5814	5971	5689
	E52	1246	2550	3936	5150	5761	5864	5618
	E53	1186	2525	3900	5089	5721	5861	5518
	E54	1182	2507	3789	5029	5650	5789	5554
	E55	1150	2404	3257	4618	5193	5261	5018
	E56	1286	2689	4097	5254	5975	6096	5731
	Average strain(FSG)	**1214**	**2511**	**3807**	**5064**	**5686**	**5807**	**5521**
	S5	**1221**	**2525**	**3825**	**5079**	**5729**	**5829**	**5557**

**Table 8 sensors-18-04106-t008:** *σ*_l2_ and *σ*_l1_ calculated from the strains of the LFBG sensors at Step I. (Unit: MPa).

Itemized Losses	*σ* _l2_						*σ* _l1_
*P*/kN	120	240	360	480	540	555	426.6
0 *	-	-	-	-	-	-	305.7
R1	20.8	32.2	37.2	34.2	45.0	53.8	225.0
R2	22.2	38.6	45.8	47.8	62.8	71.0	162.2
R3	29.2	48.6	60.0	78.6	93.6	108.8	143.6
R4	36.4	55.8	76.4	103.6	120.8	138.8	121.4
R5	40.8	65.0	90.0	124.2	139.2	155.2	54.4

* “0” is the zero point in the coordinate that represents the right end of the beam in Figure 8a.

**Table 9 sensors-18-04106-t009:** The results of strain measurement from various sensors at Step II. (Unit: με).

	Time/Hour	1	2	3	12	24	48	72
RR1	E11	5130	5111	5102	5052	5033	4999	4976
	E12	5016	4988	4976	4923	4895	4862	4853
	E13	4713	4687	4672	4609	4570	4544	4542
	E14	4888	4868	4853	4790	4777	4752	4738
	E15	5038	5012	5005	4955	4927	4899	4876
	E16	5129	5097	5090	5021	4993	4955	4949
	Average strain(FSG)	**4985**	**4960**	**4949**	**4891**	**4865**	**4835**	**4822**
	S1	**5141**	**5121**	**5113**	**5056**	**5026**	**4987**	**4970**
RR2	E21	5450	5423	5411	5338	5329	5302	5275
	E22	5348	5322	5309	5256	5231	5202	5183
	E23	5182	5156	5137	5085	5057	5030	5006
	E24	5068	5048	5039	4980	4963	4932	4905
	E25	5362	5332	5317	5272	5256	5218	5197
	E26	5402	5382	5377	5313	5293	5252	5248
	Average strain of E21–E26	**5302**	**5277**	**5266**	**5208**	**5189**	**5156**	**5136**
	S2	**5366**	**5340**	**5324**	**5251**	**5223**	**5198**	**5184**
RR3	E31	5407	5390	5377	5319	5310	5282	-
	E32	5319	5295	5276	5245	5209	5180	5166
	E33	5184	5162	5143	5097	5076	5044	5022
	E34	5114	5097	5091	5041	5017	4986	-
	E35	5175	5154	5144	5100	5082	5049	5029
	E36	5377	5358	5350	5298	5287	5266	5246
	Average strain of E31–E36	**5263**	**5243**	**5230**	**5183**	**5164**	**5135**	**5116**
	S3	**5266**	**5241**	**5231**	**5167**	**5155**	**5131**	**5118**
RR4	E41	5309	5283	5270	5210	5174	5151	5126
	E42	5321	5290	5281	5218	5181	5155	5121
	E43	5035	5011	5004	4946	4907	4883	4848
	E44	4991	4977	4964	4909	4876	4851	4825
	E45	5184	5152	-	-	-	-	-
	E46	5484	5454	5431	5371	5346	5319	5271
	Average strain of E41–E46	**5221**	**5195**	**5190**	**5131**	**5097**	**5072**	**5038**
	S4	**5215**	**5191**	**5180**	**5108**	**5070**	**5021**	**4999**
RR5	E51	5580	5544	5529	5451	5430	5392	5375
	E52	5519	5507	5491	5447	5411	5380	5345
	E53	5422	5406	5387	5310	5281	5252	-
	E54	5454	5423	-	-	-	-	-
	E55	4914	4878	4869	4808	4771	4736	4712
	E56	5631	5592	5565	5562	5544	5504	5479
	Average strain of E51–E56	**5420**	**5392**	**5368**	**5316**	**5287**	**5253**	**5228**
	S5	**5459**	**5432**	**5407**	**5363**	**5323**	**5271**	**5239**

**Table 10 sensors-18-04106-t010:** The sum of *σ*_l4_ and *σ*_l5_ calculated from the strains obtained from the LFBG sensors. (Unit: MPa).

Itemized Losses	*σ* _l4_ *+ σ* _l5_						
Time/hour	1	2	3	12	24	48	72
0 *	17.3	21.7	26.1	35.7	42.1	45.3	46.3
R1	14.0	18.0	19.6	31.0	37.0	44.8	48.2
R2	14.6	19.8	23.0	37.6	43.2	48.2	51.0
R3	15.4	20.4	22.4	35.2	37.6	42.4	45.0
R4	17.8	22.6	24.8	39.2	46.8	56.6	61.0
R5	19.6	25.0	30.0	38.8	46.8	57.2	63.6

* “0” is the zero point in the coordinate that represents the right end of the beam in Figure 8a.

**Table 11 sensors-18-04106-t011:** The monitored strains of the strands at tensioning. (Unit: με).

	Applied Stress/MPa	358.4	728.0	1023.7	862.3
R1	S11	2456	4938	6755	6196
	S12	1320	3067	4254	3606
	S13	1428	2941	4325	3972
	S14	1621	3461	4988	4298
	S15	1913	3911	5495	4805
	S16	2066	3638	5091	4611
	Average strain of S11–S16	1801	3659	5151	4581
R2	S21	2361	4705	6568	6561
	S22	1074	2532	3643	3628
	S23	759	1771	3003	2977
	S24	1945	3832	5162	5151
	S25	1888	3609	5152	5134
	S26	2027	3898	4949	4931
	Average strain of S21–S26	1676	3391	4746	4730

**Table 12 sensors-18-04106-t012:** *σ*_l2_ and *σ*_l1_ calculated from the strains of the LFBG sensors. (Units: MPa).

Itemized Prestress Losses		*σ* _l2_			*σ* _l1_
Applied stress		358.4	728.0	1023.7	862.3
R1	w1	−120.5	−234.9	−293.5	109
	w2	101.0	129.9	194.2	126.3
	w3	79.9	154.5	180.3	68.9
	w4	42.3	53.1	51.0	134.6
	w5	−14.6	−34.6	−47.8	134.5
	w6	−44.5	18.4	31.0	93.6
	Average(LFBG) *	**7.3**	**14.4**	**19.2**	**111.2**
	Prediction(Code) **	**7.9**	**16.0**	**22.5**	**108.4**
R2	w1	−102.0	−189.5	−257.1	1.4
	w2	149.0	234.3	313.3	2.9
	w3	210.4	382.7	438.1	5.1
	w4	−20.9	−19.2	17.1	2.2
	w5	−9.8	24.2	19.1	3.5
	w6	−36.9	−32.1	58.6	3.6
	Average(LFBG) *	**31.6**	**66.7**	**98.2**	**3.1**
	Prediction(Code) **	**29.4**	**59.7**	**83.9**	**0**

* represents the average stresses or prestress losses calculated from strains obtained from the LFBG sensors. ** the values are estimated based on the Chinese Code [47].

**Table 13 sensors-18-04106-t013:** The monitored strains of the strands at the in-service stage. (Unit: με).

Time	R1							R2						
	W1	W2	W3	W4	W5	W6	Average	W1	W2	W3	W4	W5	W6	Average
0	6196	3606	3972	4298	4805	4611	4581	6561	3628	2977	5151	5134	4931	4730
12 h	6052	3461	3875	4137	4681	4409	4436	6343	3454	2833	4961	4990	4800	4564
24 h	6031	3443	3857	4121	4663	4387	4417	6329	3436	2815	4950	4973	4782	4548
36 h	6020	3434	3838	4116	4654	4363	4404	6321	3427	2806	4937	4964	4771	4538
48 h	6011	3425	3819	4112	4645	4344	4393	6315	3418	2797	4923	4955	4766	4529
9 days	5959	3356	3780	4069	4582	4280	4338	6252	3353	2749	4876	4904	4714	4475
30 days	5862	3225	3717	3983	4496	4183	4244	6154	3236	2636	4776	4813	4626	4374
51 days	5810	3163	3695	3953	4459	4164	4207	6119	3209	2606	4718	4773	4599	4337
72 days	5794	3151	3686	3933	4433	4152	4192	6104	3194	2593	4694	4760	4582	4321
90 days	5784	3143	3683	3923	4422	4146	4183	6098	3181	2589	4681	4758	4570	4313

**Table 14 sensors-18-04106-t014:** The sum of *σ*_l4_ and *σ*_l5_ calculated from the strains obtained from the LFBG sensors. (Units: MPa).

Time	R1							R2						
	W1	W2	W3	W4	W5	W6	Average	W1	W2	W3	W4	W5	W6	Average
12 h	28.1	28.3	18.9	31.4	24.2	39.4	28.4	42.5	33.9	28.1	37.1	28.1	25.5	32.5
24 h	32.2	31.8	22.4	34.5	27.7	43.7	32.1	45.2	37.4	31.6	39.2	31.4	29.1	35.7
36 h	34.3	33.5	26.1	35.5	29.4	48.4	34.5	46.8	39.2	33.3	41.7	33.2	31.2	37.6
48 h	36.1	35.3	29.8	36.3	31.2	52.1	36.8	48.0	41.0	35.1	44.5	34.9	32.2	39.3
9 days	46.2	48.8	37.4	44.7	43.5	64.5	47.5	60.3	53.6	44.5	53.6	44.9	42.3	49.9
30 days	65.1	74.3	49.7	61.4	60.3	83.5	65.7	79.4	76.4	66.5	73.1	62.6	59.5	69.6
51 days	75.3	86.4	54	67.3	67.5	87.2	73.0	86.2	81.7	72.3	84.4	70.4	64.7	76.6
72 days	78.4	88.7	55.8	71.2	72.5	89.5	76.0	89.1	84.6	74.9	89.1	72.9	68.1	79.8
90 days	80.3	90.3	56.4	73.1	74.7	90.7	77.6	90.3	87.2	75.7	91.7	73.3	70.4	81.4

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
