# Peer review of "Evaluation of Prestress Loss Distribution during Pre-Tensioning and Post-Tensioning Using Long-Gauge Fiber Bragg Grating Sensors"

_sensors, 2018, doi:10.3390/s18124106_

Round 1
Reviewer 1 Report
This paper presents an innovative and more accurate method to monitor the stresses in prestressing strands in a long-term period that solves of the problems encountered in other techniques available to this end. It presents an advance in the state of the knowledge in the monitoring techniques and summarizes results that are of interest for the readers. It deserves publication as far as the following minor corrections are included:
- In lines 224-225: "The post-tensioning system overcomes a remarkable limitation of the pre-tensioning system: it enables the tendons to keep straight before and after tensioning." In my opinion, this sentence is wrong, as in postensioning the cables are never straight, either before or after tensioning. Please revise.
Figures 7 c and d are not clear enough. A zoom of 7c showing more detail is necessary and explanation of the different parts shown in figure 7d too.
In line 424 ".....and the second number represents the number of the wire". Wire should be replaced by strand.
100 days of monitoring is too short time for a full development of creep and relaxation. Therefore, I suggets the authors to enlarge the time frame of the results in figure 11 and the corresponding tables if they have more data available at the time of revising the paper.
Author Response
Response to Reviewer 1 Comments
At first, we want to appreciate these useful suggestions from the reviewers. Based on their comments, we have revised our manuscript in detail. And English of this paper has been carefully improved to enable readers to understand the proposed calculation method and the monitoring more easily in evaluation of the prestress losses.
Point 1: In lines 224-225: "The post-tensioning system overcomes a remarkable limitation of the pre-tensioning system: it enables the tendons to keep straight before and after tensioning." In my opinion, this sentence is wrong, as in postensioning the cables are never straight, either before or after tensioning. Please revise.
Response 1:Thank you for your suggestion. The sentence in lines 227 has been revised as
A remarkable limitation of the pre-tensioning system is that the tendons always have to be straight. However, the post-tensioning system enables the tendons to keep a curved profile before and after tensioning.
Point 2: Figures 7 c and d are not clear enough. A zoom of 7c showing more detail is necessary and explanation of the different parts shown in figure 7d too.
Response 2: Figures 7 c and d are replaced by the new pictures as follows. And the Figures 7 e is added to give more details.
c)
d)
e)
Figure 7. The sketches and photographs of the test design: c) the photograph of the strand attached with sensor; d) the photograph of tensioning and e) the photograph of vertical load applied on the cement-filled steel tube.
Point 3: In line 424 ".....and the second number represents the number of the wire". Wire should be replaced by strand.
Response 3: The sentence has been revised as “The first number means the sensor is in R1, and the second number represents the number of the strand.”
Point 4: 100 days of monitoring is too short time for a full development of creep and relaxation. Therefore, I suggets the authors to enlarge the time frame of the results in figure 11 and the corresponding tables if they have more data available at the time of revising the paper.
Response 4: Thank you for your suggestion. It needs a long time to observe the effect of creep and relaxation indeed. The monitoring began about July 12nd, 2018. The data shown in the paper is from that time to about Oct 15th, 2018. The authors will try our best to obtain the subsequent data to evaluate the prestress losses caused by creep and relaxation. If we accumulate enough data, they will be shown to the reader in the future paper affirmatively.

Reviewer 2 Report
Good paper. Well explained.
1. Minor english style issues such as:
Line 17: "...and so on." this can simply be stated as "... environment and vulnerable ..."
2. Line 190: do you mean mid-span instead of mid-plan?
3. Line 101: what about the mismatch in moduli between the epoxy and the steel? does not produce separation between the sensor and the test article?
Author Response
Response to Reviewer 2 Comments
At first, we want to appreciate these useful suggestions from the reviewers. Based on their comments, we have revised our manuscript in detail. And English of this paper has been carefully improved to enable readers to understand the proposed calculation method and the monitoring more easily in evaluation of the prestress losses.
Point 1:1. Minor english style issues such as:
1. Line 17: "...and so on." this can simply be stated as "... environment and vulnerable ..."
2. Line 190: do you mean mid-span instead of mid-plan?
3. Line 101: what about the mismatch in c between the epoxy and the steel? does not produce separation between the sensor and the test article?
Response 1:Thank you for your suggestion.
1. Line 17. This sentence has been revised as “The sensors attached on strand inside the duct also have several problems, such as inadequate durability in an aggressive environment and vulnerable damage at tensioning”.
2. Line 193 “...region near the mid-plan of the beam...” is revised as “...region near the mid-span of the beam...”
Line 195 “The reason for the latter is that the mid-plan usually...” is revised as “The reason for the latter is that the mid-span usually...”
3. According to the theory of mechanics of materials, if the bond is undamaged, the tensile deformation of epoxy equals to that of the steel strand. However, due to the mismatch in moduli between the epoxy and the steel, the shear stress is limited. It means that the transformed tensile stress is small and cannot meet the requirement to tension the sensor encapsulated by steel tube. The elastic module of fiber reinforced polymer (FRP) is remarkable smaller than that of steel. Thus, if the sensor is packaged by epoxy-soaked fiber reinforced polymer (FRP) material, the elongation of the sensor is similar to that of steel strand. It will not produce separation between the sensor and the test article basically. The tests in the paper testify this conclusion.
